# Design of SiC-Doped Piezoresistive Pressure Sensor for High-Temperature Applications

**DOI:** 10.3390/s21186066

**Published:** 2021-09-10

**Authors:** Tomasz Wejrzanowski, Emil Tymicki, Tomasz Plocinski, Janusz Józef Bucki, Teck Leong Tan

**Affiliations:** 1Faculty of Materials Science and Engineering, Warsaw University of Technology, Woloska 141, 02 507 Warsaw, Poland; tomasz.plocinski@pw.edu.pl (T.P.); janusz.bucki@pw.edu.pl (J.J.B.); 2Łukasiewicz Research Network—Institute of Microelectronics and Photonics, Al. Lotników 32/46, 02-668 Warsaw, Poland; emil.tymicki@itme.edu.pl; 3ENSEMBLE3 Sp. z o.o., Wolczynska Str. 133, 01-919 Warsaw, Poland; 4Institute of High Performance Computing, Agency of Science, Technology and Research (A*STAR), 1 Fusionopolis Way, #16-16 Connexis, Singapore 138632, Singapore; tantl@ihpc.a-star.edu.sg

**Keywords:** SiC, piezoresistive effect, high temperature pressure sensor, gauge factor

## Abstract

Within these studies the piezoresistive effect was analyzed for 6H-SiC and 4H-SiC material doped with various elements: N, B, and Sc. Bulk SiC crystals with a specific concentration of dopants were fabricated by the Physical Vapor Transport (PVT) technique. For such materials, the structures and properties were analyzed using X-ray diffraction, SEM, and Hall measurements. The samples in the form of a beam were also prepared and strained (bent) to measure the resistance change (Gauge Factor). Based on the results obtained for bulk materials, piezoresistive thin films on 6H-SiC and 4H-SiC substrate were fabricated by Chemical Vapor Deposition (CVD). Such materials were shaped by Focus Ion Beam (FIB) into pressure sensors with a specific geometry. The characteristics of the sensors made from different materials under a range of pressures and temperatures were obtained and are presented herewith.

## 1. Introduction

The piezoresistive effect has been known since 1856 thanks to the discovery of Lord Kelvin (William Thomson) [1]. It describes the electrical resistivity change in a material in response to mechanical deformation. This property is frequently quantified by the so-called gauge factor (GF), which represents the fractional change in the resistance per unit strain:(1)$$GF=\frac{\Delta R}{R}\frac{1}{\varepsilon }$$
where R is the resistance and ε indicates strain.

GF is dimensionless and is positive when the resistance increases with increasing tensile strain and is negative when resistance decreases with increasing strain.

The piezoresistive effect is widely used to convert a mechanical loading into an electrical signal. The effect is utilized in different devices such as pressure sensors [2,3,4,5,6,7,8,9,10,11,12,13], tactile sensors [14], strain gauges [15], flow sensors and others [16,17,18,19]. Piezoresistive materials are characterized by their unique electromechanical coupling and have recently received growing interest with the miniaturization of electromechanical devices down to micro or even nano scales. Novel material technologies not only enable device miniaturization, but also lead to enhanced sensitivity, lower cost, and higher stability in harsh environments. Apart from the materials used, the construction of the sensor and the type of electrical circuit play a key role for the sensitivity and in particular, integration of the sensor with other microelectronics. The most popular electrical circuit is the Wheatstone bridge type, which enables resistance tuning and noise reduction. New, more sophisticated systems are also capable of compensating for nonlinearity [20] as well as sensitivity changes with temperature [21]. In addition, there are a variety of different sensor geometries in the literature. In particular, the strain distribution along the membrane/diaphragm is one of the crucial elements to optimize [22,23]. The placement of piezoresistive elements on the membrane as well as their geometry were also studied in order to maximize sensor sensitivity [24].

To date, most of the piezoresistive devices and in particular, pressure sensors have been manufactured based on silicon (Si) technology, taking advantage of its physical and chemical properties along with its mature fabrication technologies [2,25]. Many investigations have also been made on low-dimensional nanostructures, such as Si nanostructures [26,27], ZnO nanostructures [28,29], Si_3_N_4_ [30,31], carbon nanotubes [32,33], and graphene [34,35,36], for their excellent piezoresistive effect and inspired potential applications.

However, most of the devices based on Si and the abovementioned materials, are often designed for operations at room temperatures, which greatly limits their use in harsh environments, particularly under high temperature conditions. This drawback can be overcome by the use of silicon carbide (SiC), which has a wider bandgap, higher thermal conductivity, higher Young’s modulus and a lower thermal expansion coefficient than Si. For instance, the SiC pressure sensors were confirmed to perform well up to 700–800 °C [8,13].

The comprehensive review of various materials and their applications have been presented by HP Phan at al. [4]—see Figure 1. The study indicates that SiC-based devices are much more favorable at higher temperatures and less applicable at room temperatures where doped Si presents a 5–6 times stronger piezoresistive effect (gauge factor).

SiC exists in many different crystal structures, which are called polytypes [37]. There are more than 200 SiC polytypes, but most studies and developments have focused on three types: 3C-, 4H-, and 6H- SiC. Among them, the 4H- and 6H- polytypes are more favorable for piezoresistive devices owing to their excellent properties [38]. The energy band gap of 4H-SiC is 3.2 eV and 3.0 eV for 6H-SiC which is considerably higher than that of 3C (2.3 eV). The high potential barrier in SiC materials can effectively minimize the number of electron-hole pairs generated at high temperatures across the band gap, enabling the high temperature stability of SiC electronic devices and sensors [38,39,40].

The change in resistance arises from two effects: the change in the dimension and geometry of the resistor and the change in the resistivity of the material itself. The piezoresistance of a material is affected by many factors. Apart from chemical composition and crystal structure (polytype), the crystal orientation along with the dopant type and its concentration play important roles.

For SiC polytypes the effect of various dopants on the gauge factor have been studied. In particular, materials with N, B, and Al doping were fabricated to obtain n- or p-type conductivity. Selection of such dopants were made based on experimental studies, which reveal that the SiC crystal structure remains stable over a wide range of dopant concentrations. The properties of SiC polytypes with various dopants are shown in Table 1.

The results presented in Table 1 indicate that usually for n-type SiC, a negative gauge factor is expected, where its resistivity is reduced with an increase in strain. For p-type SiC, the GF value is typically positive. Reduction in the sensitivity is observed with increasing temperature. Dopant concentration is crucial for tuning the electronic conductivity of semiconductor devices. On the other hand, the experimental results indicate that GF is significantly reduced for larger dopant concentrations. Taking those two aspects into consideration, the optimum concentration is found to be 10^18^–10^19^ cm^−3^ for nitrogen [4] and one order of magnitude lower for boron dopant [56], respectively.

Recently, about a hundred different substitutional elements were explored using Density Functional Theory (DFT) calculations [57] to improve the GF of SiC. Among them, Ru substitution of C-sites with a concentration of 1.56 at% improved gauge factor by around four times. Pt and Pd at Si-sites increased GF at higher temperature. These DFT predictions need experimental verification of the GF and the stability of the SiC crystal structure after doping.

Typical pressure sensor devices are made up of a few functional layers; the piezoresistive layer is deposited on the substrate and coated with an electronic contact layer. To ensure good adhesion between functional layers, very often some diffusion interlayers are formed between them. Two manufacturing technologies are dominant: heteroepitaxy of piezoresistive SiC on Si substrate [5] and homoepitaxy of doped SiC on undoped SiC substrate [58]. For the contact layer, Al or Au thin films are usually used. Each technology is dedicated to specific application, however, homoepitaxy of SiC provides the only solution for high temperature use.

Pure SiC crystal used for substrates are usually fabricated by the Physical Vapor Transport (PVT) method [59]. A thin piezoresistive layer can be grown on the substrate by various Chemical Vapor Deposition techniques. Doping is realized by modification of the reactive gas compositions. Focused Ion Beam (FIB) is frequently used [26,60,61] to fabricate piezoresistive devices at the design stage, whereas in commercial application, optimized devices are shaped by lithography [50].

It has been shown that the thickness of the piezoresistive layer influences GF. For the same device dimensions, the thinner the layer the higher is the resistance, which in turn could lead to a higher GF. This is attributed mostly to a more favorable strain distribution at smaller dimension systems, resulting in a larger average strain due to accumulated tensile loading when the piezoresistive thin film bends. Thus, to reduce the overall resistivity and retain a high sensitivity, the sensor dimension is reduced to sub-millimeter size. The geometry of the whole device is also important for optimizing the transfer of applied strain to the piezoresistive layer, where the substrate is shaped to form a cantilever or membrane with the specific stiffness.

In this work, we fabricated and tested at room and elevated temperatures, pressure sensors based on the most promising 4H-SiC polytype modified by various dopants: N, B, and Sc. The unique geometry of the sensors enables measurement and compensation of the anisotropic character of the piezoresistive effect and was designed via numerical simulation before fabrication using Focused Ion Beam. We provided characteristics for a specific sensor geometry to enable further design of the devices dedicated to specific pressure range measurements.

## 2. Materials and Methods

In the first stage of our investigation, bulk 4H- and 6H-SiC crystals with various dopant type and concentration were fabricated by PVT and characterized with respect to their structure and electronic properties. The piezoresistance and GF of bulk samples were analyzed using specially designed test stages.

Based on the results obtained for bulk crystals the most promising dopant types and concentrations were selected and used to produce thin-film 4H- and 6H-SiC via CVD deposition on undoped SiC substrate. The layered structures were characterized and used to fabricate piezoresistive element shaped by FIB. In the last stage, pressure sensors were constructed and tested in the pressure range of 0–5 bar and temperature up to 300 °C by using a specially developed testing rig.

### 2.1. Bulk Crystals Manufacturing and Characterization

Bulk SiC crystals were grown using the PVT method with the experimental setup equipped with two resistive graphite heaters. The furnace was also equipped with a gas system that allowed crystallization in the mixed atmosphere of argon and nitrogen, with different N_2_/Ar ratio. A schematic of the growth chamber is shown in Figure 2a.

The source material was 500 g of SiC powder with grain size of ~0.1 mm. The source material was initially purified by annealing for 30 h at 2300 °C under 20 mbar pressure of Ar. Solid dopants, such as boron and scandium, were introduced into the growth system by filling a graphite container with powder (purity of 99.999%), placed in the middle of the SiC source material (see Figure 2b). A small vent was made at the top of the container to allow for slow vaporization of aluminum or boron into the growth atmosphere.

Crystals were grown using on-axis SiC crystal seeds, 2” in diameter and 1 mm thick. The orientation of the seed is a crucial factor for polytype stabilization. According to the literature, only seeds with the C-face polarity promote stable growth of the 4H-SiC polytype, hence our growth processes were performed on the (000-1) oriented seeds. The seeds were installed in the experimental setup via the open backside method [62].

During the growth, the temperature measured at the backside of the seed was 2150 °C. The temperature of the source material was set at 2200 °C. The total pressure inside the growth chamber was kept steady at 40 mbar. The growth processes were carried out for a period of 50 h for each crystal.

During the first stage of the processing, the side surfaces of the SiC crystals were grounded. The purpose of this operation was to remove the edge area and obtain a cylindrical shape. In the next stage, the base was milled in the *m* direction [1-100] and a mark was made in the *a* direction [11-20], along the side surface of the crystal. This ensured the crystallographic orientation and marking on the carbon and silicon faces of the cut SiC wafers. The cutting process was carried out using a circular saw with a 0.4 mm thick blade. The thickness of the cut wafers was 1 mm. The wafers were then sanded with use of a diamond suspension with grain size of 16 µm. This was followed by pre-polishing and finishing. Initial polishing was carried out on a canvas in a diamond suspension with a grain diameter of 6 µm while the final polishing was performed in a suspension with a grain diameter of 1 µm. The roughness, Ra, of the wafer surface was below 5 nm. Specimens of specific shapes were cut from the polished wafers on both sides according to the needs of each specific research method (see Figure 3a). The samples with the final thickness were cut into smaller elements (1 × 1 cm) and characterized (see Figure 3b).

The morphology and the crystalline structure of SiC growth surfaces were studied by optical microscopy (before and after etching) and X-ray diffraction. Analysis of crystal defect density (micropipes) was performed by analysis of optical images using the software developed at Warsaw University and Technology—SDA Semiconductor Defect Analyzer.

The concentration of dopants and resistivity were evaluated by the Hall method. Within this technique, measurements of electrical parameters such us ρ (Ωcm), n (cm^−3^), µ (cm^2^/Vs), and the type of conductivity were performed at room temperature and liquid nitrogen temperature (77 K) using the Van der Pauw constant current method. Samples with sizes of 10 × 10 mm and 500 µm thickness with four contacts in the corners were tested with a laboratory measuring system set up from Keithley meters. The magnetic field during the Hall measurements was 1.5 T.

The Raman scattering was chosen as a non-contact and non-destructive characterization method for analyzing SiC polytypes for both the lattice and electronic properties. Raman scattering efficiency of SiC is high because of strong covalency in the chemical bonding, and one can easily extract various kinds of useful information on SiC such as polytype, defects, lattice strain, impurities, free carrier density, and mobility [63]. Micro-Raman scattering experiments were performed at room temperature with backscattering geometry, using the 532 nm line from Nd-YAG as a source of continuous wave excitation. Laser spot size on the sample surface was about 2 µm.

Measurements of the piezoresistive effect were performed using a 4-contact beam that was deformed during the measurement as shown in Figure 4. The 4-point circuit provided voltage measurement across the sample without voltage drop across the sample current leads. This type of connection allows for a more precise measurement of the voltage drop, and thus sample resistance as a function of deformation. The beam was made of monocrystalline silicon with a width of 10 mm and a thickness of 500 µm. The tested SiC system was attached to a beam using an adhesive (two-component epoxy glue PRO WELD^®^ QUICK Pro Seal) with a low elasticity factor. These solutions enabled stress transmission from the silicon beam to the SiC system. The deformation was caused by using a micrometer screw, and the voltage drop was measured using a microvoltmeter. The strain, ε, in the piezoresistive element was calculated based on Equation (2) [64]:(2)$$\varepsilon =\frac{3fh}{2l^{2}}$$
where *f* denotes deflection arrow (deformation induced by micrometer screw), *h* and *l* are a cantilever height and length, respectively.

### 2.2. Layered Materias Fabrication and Characterization

The epitaxial layers were grown using the horizontal hot-wall chemical vapor deposition (CVD) method with reactor Epigress VP508GFR. The growth temperature was in the range 1600–1650 °C and the reactor pressure was 80 mbar, along with 50l/min of H_2_ as carrier gas. Propane (C_3_H_8_) and silane (SiH_4_) gases were used as carbon and silicon precursors, respectively. Before the epitaxial growth, in situ H_2_ pre-etching was done to remove the damaged layer containing scratches. During the growth, a gaseous source of dopants was intentionally added to the gas mixture. Samples were cut and prepared the same way as used for the bulk materials studied. For the characterization purpose 15 × 15 mm square samples were cut from polished wafers. The thickness of the epitaxial layers was measured under an optical microscope with Nomarski contrast. For the sensor construction 6 × 1 × 0.070 mm (length x width x height) beams were cut and coated with a thin film of gold (about 4 µm thick)—see Figure 5. To enhance the adhesion of gold film to SiC, the chromium interlayer was deposited.

Depth concentration of dopants in layered materials was analyzed by Secondary Ion Mass Spectrometry (SIMS). The SIMS method is based on the detection of atoms and molecules knocked out from the sample surface by a beam of high energy ions. A cesium cannon bombarding 133Cs atoms was used to analyze the samples. This method allows determination of the concentrations of individual elements within the sample. The measurements were performed on the CAMECA SIMS IMS6F spectrometer.

### 2.3. Design, Fabrication, and Characterization of Pressure Sensors

The geometry of the piezoresistive element and the construction of the sensor was designed (see Figure 6) using SolidWorks software (version 2019); the piezoresistive element was produced via cutting of the sample using Focused Ion Beam (FIB).

A 50 µm thick 316-steel plate was used as a membrane. The SiC beam with piezoresistive element was placed on the membrane and fixed at its ends with high temperature silicone. The ends of the beam were also fixed mechanically due to cutouts in the brass pads pressed by the sensor case. The system was numerically simulated to analyze strain distribution under a pressure of 0–5 bar.

Fabrication of piezoresistive elements were performed by using FIB model NB5000 made by Hitachi High Technologies. This system is a dual beam microscope, equipped with SEM and FIB columns. Fabrication using FIB is not the most effective way to produce the sensors because of the long manufacturing time and high beam cost, but at the design stage this technique is the most flexible one, allowing quick changes to the prototype sensor. The fabrication of the grooves was conducted in two steps. First, sputtering was carried out at a maximum ion beam current of 55 nA. To obtain the best shape for the grooves, several different scanning strategies of the ion beam were explored. The best results were obtained when alternating scanning with 5 ms dwell time was applied. To remove the conductive layer of gold from the specific area of the sensor, a 15 nA ion beam was used. After fabrication, the final structure was checked under SEM, using 5 kV accelerating voltage. For observation, the SE (secondary electron) signal was used. The atomic contrast remains visible on the pictures, which was essential in distinguishing how well the gold layer was removed during the final sputtering. An example of the sensing element cut by FIB is shown in Figure 7.

The electronic contacts were made using thin gold wire (diameter 15 µm), which was bonded to the square fields prepared by FIB.

In order to analyze the piezoresistive effect of the fabricated sensors, a special testing bench was constructed as shown in Figure 8.

The main part contained the chamber in the form of a pipe where two sections could be distinguished—the cold and hot zone. In the cold zone, the air was supplied by the compressor to adjust the pressure, which was controlled by an electronic high precision manometer (BD Sensors DM01). The pressure sensors were mounted in the hot zone heated by a micanite heater ring. A full-PID mode temperature controller equipped with thermocouple (REX C-700+ K type probe) was used to stabilize the temperature of the sensor at 28, 100, 200, and 300 °C. The hot zone was separated from the cold zone by copper pipe coils with flowing water. 

Electrical wiring of the sensor (thin gold wires) was connected to the printed circuit board located in the cold zone. A 4-point method was used to measure the resistance of the sensor. The electrical resistance of the sensor at the specified temperature (varied from 28 to 300 °C) and the pressure (from 0 to 5 bar) was calculated from the linear relationship between the voltage supplied and the current measured. The resistivity was obtained from the resistance, taking into account the geometry of the sensing element obtained by SEM imaging.

## 3. Results and Discussion

### 3.1. STRUCTURE and Properties of Doped Bulk SiC

For the 2-inch wafers, the estimated average micropipe density for each fabricated sample was not larger than *n* = 2 cm^−2^ (see Figure 9). It is worth mentioning that after etching with KOH, the density of pits ranged from 10^4^ to 10^5^ cm^−2^. Such a relatively low density of micropipes and dislocations indicates the high quality of the materials fabricated.

In order to identify SiC polytypes, the XRD patterns of powder samples (produced by crushing the 5 × 5 × 1 mm specimens) were analyzed. The results are presented in Figure 10, where it is evident that experimental diffraction peaks are almost 100% in agreement with the patterns of the 4H- and 6H-SiC polytypes. Both polytypes have their own characteristic XRD pattern. XRD peak positions corresponded to the 4H-SiC (ICDD-PDF: 29-1127) and 6H-SiC (ICDD-PDF: 29-1131). Moreover, in the 4H- and 6H-SiC samples with nitrogen or boron dopants, we did not observe change in the position of the peaks and their intensities, so the dopants such as nitrogen and boron did not change the polytype and had a negligible effect on the lattice constant. Additionally, the X-ray rocking curve (in Figure 10c) was measured for a sample that was cut from the 4H-SiC crystal. The rocking curve does not show any peak broadening, indicating that the investigated area was free of mosaic structure.

The Hall effect measurements consistently showed the n-type and p-type electrical conductivity. Samples unintentionally doped and doped with nitrogen showed n-type electrical conductivity, while samples doped with boron showed p-type electrical conductivity. The measurement was also used to determine the resistivity and mobility of the carriers. As the concentration of carriers increased, their mobility decreased, which is consistent with the literature databases [65]. The resistivity was also measured by a non-contact method using microwave radiation (MF). The obtained results of the electrical properties tests are presented in Table 2.

The characteristic Raman spectra were obtained for 4H- and 6H-SiC polytypes with various dopants (see Figure 11). The position and height of the peaks indicate homogeneity of the polytype in the samples. We notice that peaks in the 4H- and 6H-SiC spectrum are shifted to lower frequencies with increasing nominal nitrogen and boron doping concentration. A similar effect was observed by other authors [66].

From the LO+PL mode position, the concentration of free carriers was estimated for the samples 6H-SiC: 0%N (5 × 10^17^ cm^−3^), 6H-SiC: 3%N (3 × 10^18^ cm^−3^), 6H-SiC: 10%N (8 × 10^18^ cm^−3^), and for the samples 4H-SiC: 0%N (1 × 10^17^ cm^−3^), 4H-SiC: 3%N (5 × 10^17^ cm^−3^), 4H-SiC: 10%N (1 × 10^18^ cm^−3^). For the 6H-SiC: B samples, the mode changes were insufficient to estimate the carrier concentration. A shift of the low-energy mode from 151 cm^−1^ to 170 cm^−1^ was also observed for the 6H-SiC: B samples. The explanation of this phenomenon requires further research.

Bulk SiC crystals with various dopant type and concentration were tested for piezoresistivity effect by analyzing their resistivity change under strain resulting from the bent cantilever—see Figure 4. The strain in the crystal was calculated (Equation (2)) based on the geometry of the system and the deflection of the cantilever tip induced by the micrometer screw [67,68]. The GF measurements are presented in Table 2. It can be observed that the GF values for analyzed bulk materials are relatively low when compared with the literature (Table 1). This could be explained by the complex strain distribution in the bulk sample and contradictory effects for compressive and tensile strains existence. Such an effect was also studied in more detail for silicon [69], where sensitivity was improved by 120% via reduction of sensor dimensions. The maximum absolute values of GF for materials doped with nitrogen both for 4H and 6H were observed for the concentration of nitrogen in the range of 5.00 × 10^18^ cm^−3^. Further doping decreases material piezoresistance sensitivity. Doping with boron produced a positive GF. Larger concentration of boron up to 4.12 × 10^17^ cm^−3^ had no influence on the GF. Co-doping of nitrogen and scandium resulted in an increase of GF to 7.2.

The group of perspective dopants and its concentration was selected to fabricate layered materials, eventually used for the construction of a pressure sensor.

### 3.2. Pressure Sensor Based on Thin Film Doped SiC

Within these studies, four materials were selected to fabricate the layered system used to construct piezoresistive pressure sensors. The selection was based on the results of the GF measured on the bulk materials.

For the materials selected the depth concentration of the elements was measured by SIMS. The results are sown in Figure 12.

SIMS measurements showed the presence of nitrogen doping in the 4H-SiC structure, which, as predicted, increased with nitrogen doping in the growth atmosphere. Since nitrogen is a donor in SiC, these crystals were of n-type of electrical conductivity. The concentration of scandium atoms in 6H-SiC-N-Sc was estimated for the implant reference 6H-SiC sample to be below 1 × 10^18^ cm^−3^. Other unintentional dopants such as aluminum and boron were also observed. However, their concentration is at least an order of magnitude lower than the main dopant. The aluminum impurity probably came from the SiC source material and the boron impurity from the graphite elements. The boron concentration was of the same order of magnitude for the three crystals with different nitrogen doping, while the aluminum concentration increased with increasing nitrogen doping.

The piezoresistive effect was analyzed based on the resistivity changes due to the mechanical strain induced by the applied pressure. The relationship between strain and pressure for the specific sensor geometry used here was obtained by finite element modeling and is presented in Figure 13. The Gauge Factor was then calculated using Equation (1).

Each sensor was exposed to pressure to analyze the piezoresistive sensitivity (GF). In this case, the resistivity changes due to strain were measured (see Figure 14a). The trends of GF versus temperature were also analyzed (Figure 14b).

For all sensors, the linear relationship between resistivity and pressure/strain can be found. The pressure non-linearity in the range from 0–5 bar was 0.39%; 0.41%; 1.3%; 0.48% for 6H-SiC-3%N; 4H-SiC-3%N, 6H-SiC-3gB, and 6H-SiC-Sc-N, respectively. In the case of 6H-SiC polytype doped with N (6H-SiC 3%N) and Sc (6H-SiC N Sc) the resistivity increases with the increase in pressure. The opposite trend is observed for 4H-SiC polytype doped with nitrogen and 6H-SiC polytype doped with boron. In commercial sensors the overall resistivity range can be tuned by other electronics.

Despite relatively large GF, as considered for SiC sensors, the drop in GF is high when the temperature increases to 300 °C and ranges between 30–60% depending on the material. The literature review suggests that this effect is especially visible for 6H-SiC polytype with larger dopant concentration (see Table 1). Larger concentration of dopants typically increases electronic conductivity of SiC crystal, and intensifies its variation as a function of temperature, which have a negative effect on GF [4]. Reduction of the electrical resistivity with temperature rise is much higher for boron than for nitrogen [56]. Thus, we recommend the application of nitrogen as a SiC dopant when sensing stability is the critical design criterion.

For intermediate temperature sensors based on Si, such as SOI (silicon-on-insulator) [70,71] or other materials i.e., based on diamond [72], this issue is similar but less pronounced. In all the cases such deficiency requires a temperature compensation method when the sensor is going to be applied in larger temperature ranges.

## 4. Summary and Conclusions

In this study, the 4H- and 6H-SiC crystals with various dopant concentrations of nitrogen, boron, and scandium were fabricated by the Physical Vapor Transport (PVT) method. Both the structure and electronic properties were analyzed in detail by several techniques. The effects of crystal structure, dopant type and concentration on the resistance change due to mechanical strain were realized by deformation of the sample placed on the cantilever.

The results indicate that piezoresistive sensitivity described by the Gauge Factor (GF) can be increased by doping with each element used. The highest absolute values of GF were obtained for nitrogen dopant in the range of 1 to 5 × 10^18^ cm^−3^. Further increase in dopant concentration lowers the GF even though the electrical conductivity is increased for larger amounts of doping elements. The highest absolute value of GF (about 35) was obtained for co-doping of nitrogen and scandium.

The results of the extensive studies dedicated to bulk SiC crystals were utilized to fabricate and analyze layered materials, used later to construct pressure sensors. CVD homoepitaxy was applied to fabricate thin SiC doped piezoresistive films on undoped SiC substrates. Application of Focused Ion Beam (FIB) enabled the formation of micro-sensing element. The mechanical strain necessary for induction of the piezoresistive effect was realized by application of the thin 316 steel membrane. The design of the sensor geometry and prediction of strain were strongly supported by numerical simulations using the Finite Element Method.

Four sensors were constructed and tested at various conditions involving different temperatures. The results showed that the values of the Gauge Factor of the proposed materials are relatively high and remain so even at the relatively high temperature of 300 °C. Among the samples studied, we found that co-doping of 6H-SiC with nitrogen and scandium results in the highest GF (25–35), which is significantly higher than samples doped with only one element (nitrogen and boron).

## Figures and Tables

**Figure 1 sensors-21-06066-f001:**
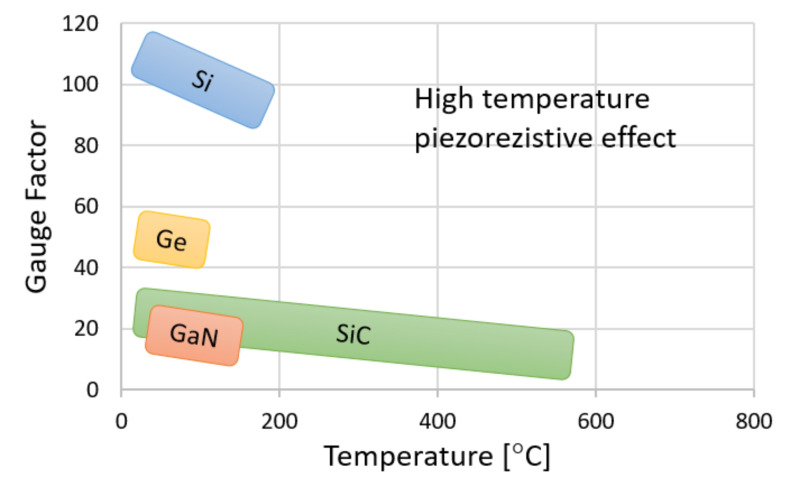
Effect of temperature on the piezoresistive gauge factor for typical semiconducting materials [4]. © 2015 IEEE.

**Figure 2 sensors-21-06066-f002:**
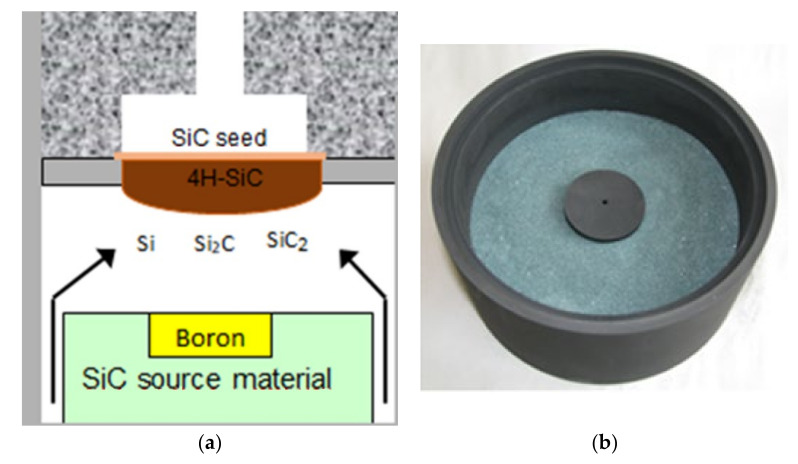
Scheme of the growth chamber for PVT crystallization of bulk materials (**a**) and SiC source in a graphite crucible (**b**).

**Figure 3 sensors-21-06066-f003:**
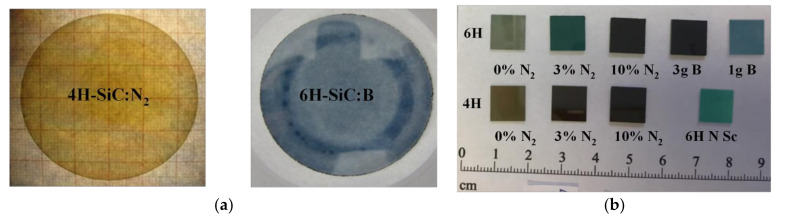
Examples of wafers cut from the bulk crystals obtained by the PVT method (**a**) and smaller samples cut from the wafers used for characterization (**b**).

**Figure 4 sensors-21-06066-f004:**
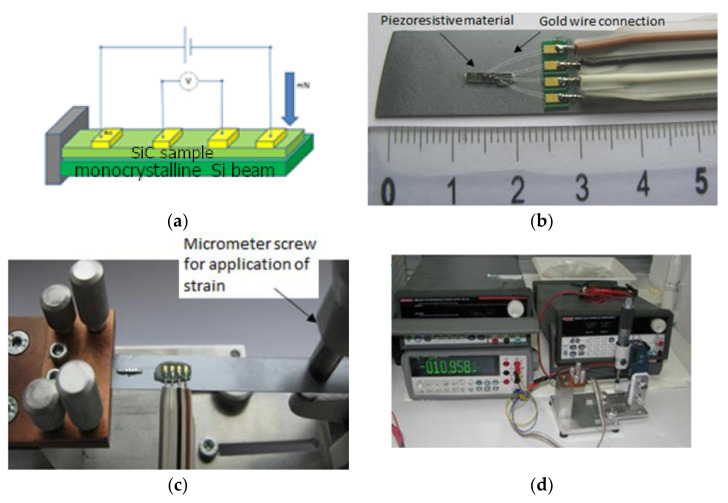
The testing stage used to analyze the piezoresistive effect of bulk crystals at room temperature. Schematic (**a**), connectors (**b**), mounting with micrometer screw (**c**) and general view of the testing stage (**d**).

**Figure 5 sensors-21-06066-f005:**
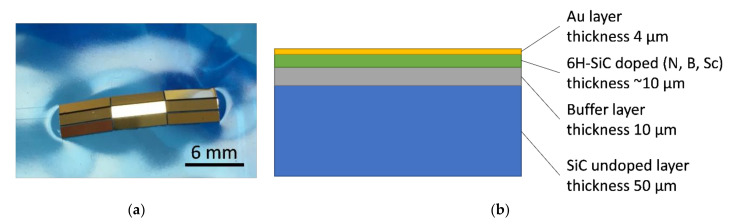
Multilayered SiC samples (**a**) and schematic illustration of layers on the cross-section (**b**).

**Figure 6 sensors-21-06066-f006:**
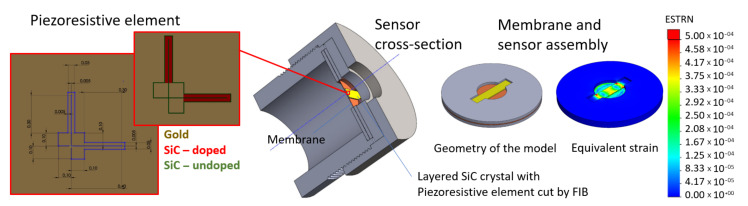
The model of the pressure sensor construction and the designed geometry of the piezoresistive element.

**Figure 7 sensors-21-06066-f007:**
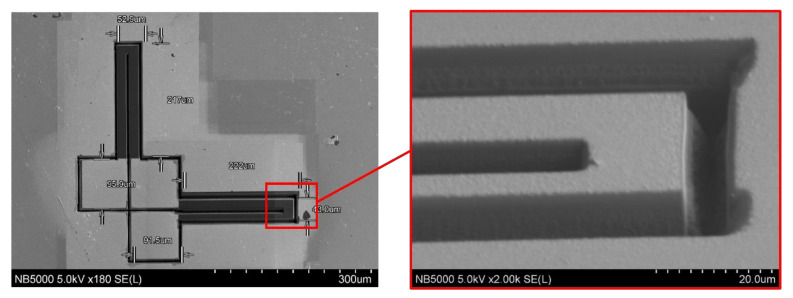
The structure of the designed sensor geometry cut by FIB.

**Figure 8 sensors-21-06066-f008:**
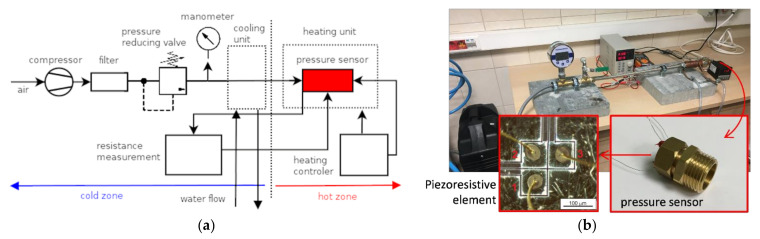
The schematic diagram (**a**) and the stage for characterization of the piezoresistive pressure sensors (**b**).

**Figure 9 sensors-21-06066-f009:**
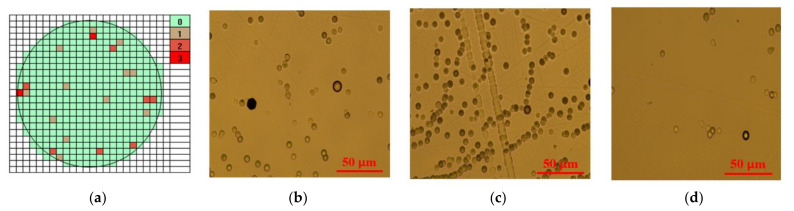
Micropipe maps of 4H-SiC wafers for SiC:N (10 vol%) (**a**) and defects at the Si face of 4H-SiC revealed after etching in molten KOH (**b**–**d**).

**Figure 10 sensors-21-06066-f010:**
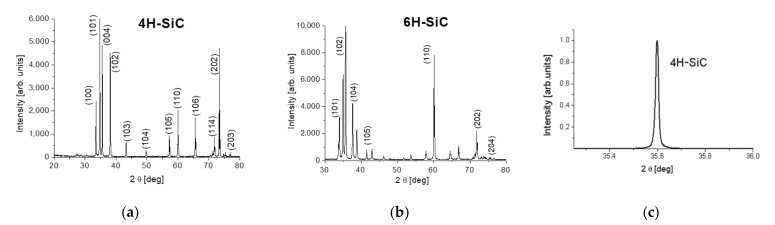
X-ray diffraction pattern for 4H- (**a**) and 6H- (**b**) SiC, and the X-ray rocking curve for 4H-SiC (**c**).

**Figure 11 sensors-21-06066-f011:**
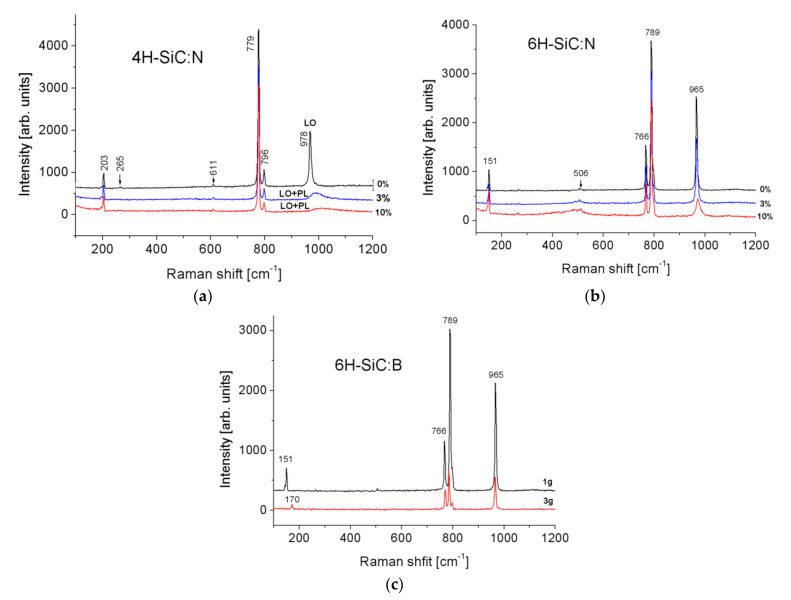
The Raman spectra of N-doped 4H-SiC (**a**) and 6H-SiC (**b**) samples and for B doped 6H-SiC sample (**c**).

**Figure 12 sensors-21-06066-f012:**
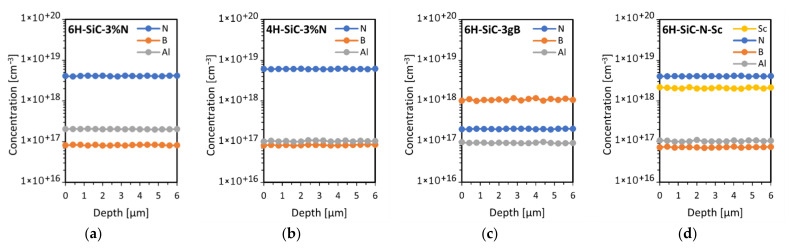
SIMS depth profiles for the studied samples: 6H-SiC-3%N (**a**), 4H-SiC-3%N (**b**), 6H-SiC-3gB (**c**) and 6H-SiC-N-Sc (**d**).

**Figure 13 sensors-21-06066-f013:**
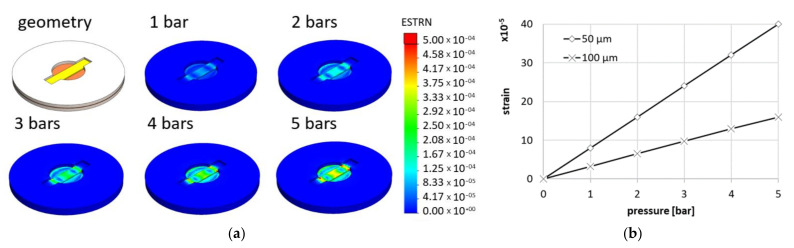
The strain of the piezoresistive element calculated for various pressures and thicknesses of the membrane –strain distribution for various pressures (**a**) and maximum strain as a function of pressure (**b**).

**Figure 14 sensors-21-06066-f014:**
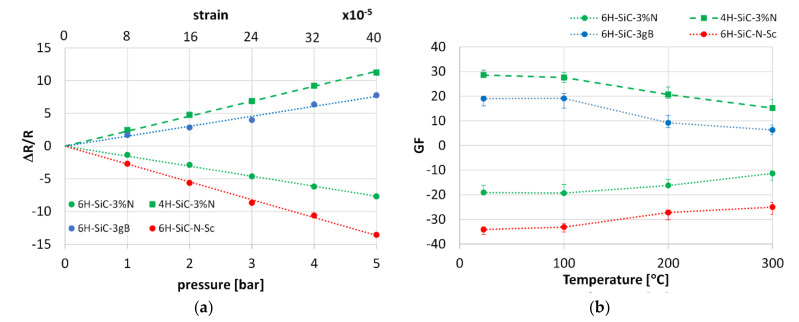
The effect of pressure on the sensor resistivity change (**a**) and Gauge Factor of sensors as a function of temperature (**b**).

**Table 1 sensors-21-06066-t001:** Effect of typical dopants on the properties of SiC based piezoresistive materials.

Polytype	Dopant Type/Element	Growth	Carrier Concentration	Gauge Factor		Orientation	Stress
				Room Temp	High Temp		
Single 4H-SiC [41]	n/N	-	1.5 × 10^19^	20.8	-	(0001)	uniaxial
Single 6H-SiC [6]	n/N	-	3.8 × 10^18^	−29.4	−17 (250 °C)	(0001)	uniaxial
Single 6H-SiC [42]	n/N	-	2 × 10^19^	−22	−11 (250 °C)	(0001)	uniaxial
Single 6H-SiC [42]	p/Al	-	2 × 10^19^	27	12 (250 °C)	(0001)	uniaxial
Single 3C-SiC [43]	n/N	APCVD	10^18^	−31.8	−18 (450 °C)	[100]	uniaxial
Single 3C-SiC [44]	n/N	HMCVD	10^18^	−27	-	[100]	uniaxial
Single 3C-SiC [45]	n/N	APCVD	-	−18	−7 (400 °C)	[100]	biaxial
Single 3C-SiC [46]	n/N	LPCVD	0.4–2 × 10^19^	−24.8	−11 (450 °C)	[100]	biaxial
Single 3C-SiC [47]	n/N	APCVD	highly doped	−16	−12.5 (400 °C)	[100]	uniaxial
Single 3C-SiC [48]	p/Al	LPCVD	5 × 10^18^	30.3	-	[110]	uniaxial
Single 3C-SiC [49]	p/Al	LPCVD	1.3–10 × 10^18^	20–30	-	[110]	uniaxial
Single 3C-SiC [50]	p/Al	LPCVD	highly doped	28	25 (300 °C)	[110]	uniaxial
Poly 3C-SiC [51]	n/N	LPCVD	low doped	−10	-	-	biaxial
Poly 3C-SiC [44]	n/N	LPCVD	-	−2.1	-	-	biaxial
Poly 3C-SiC [52]	p/B	LPCVD	10^18^–10^20^	10	7 (200 °C)	-	uniaxial
Nanocrystalline SiC [53]	p/Al	LPCVD	2 × 10^18^	14.5	-	-	uniaxial
Amorphous SiC [54,55]	n/N	PECVD	-	49	-	-	uniaxial
	n/N	Sputtering	-	31	-	-	uniaxial

N = Nitrogen, Al = Aluminum, B = Boron.

**Table 2 sensors-21-06066-t002:** The results of Hall and GF measurements for the bulk SiC crystals with various dopants.

Sample	*n* [cm^−3^]	ρ [Ωcm]	µ [cm^2^/Vs]	Type	GF
4H 0%N	5.64 × 10^17^	4.64 × 10^−2^	2.38 × 10^2^	n	2.1 ± 0.4
4H 3%N	5.00 × 10^18^	1.60 × 10^−2^	7.78 × 10^1^	n	6.4 ± 2.5
4H 10%N	9.28 × 10^18^	1.16 × 10^−2^	5.81 × 10^1^	n	4.8 ± 1.3
6H 0%N	2.34 × 10^16^	8.78 × 10^−1^	2,12 × 10^2^	n	−1.2 ± 0.3
6H 3%N	1.28 × 10^18^	3.82 × 10^−2^	1.28 × 10^2^	n	−5.8 ± 2.3
6H 10%N	6.15 × 10^18^	1.85 × 10^−2^	5.49 × 10^1^	n	−2.8 ± 0.7
6H B 1g	1.08 × 10^16^	2.39 × 10^−1^	2.41 × 10^1^	p	4.4 ± 1.6
6H B 3g	4.12 × 10^17^	3.34 × 10^−2^	1.28 × 10^1^	p	4.6 ± 1.4
6H N Sc	4.60 × 10^18^	1.81 × 10^−2^	6.12 × 10^1^	n	7.2 ± 2.6

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
