# Peer review of "Design of SiC-Doped Piezoresistive Pressure Sensor for High-Temperature Applications"

_sensors, 2021, doi:10.3390/s21186066_

Round 1

Reviewer 1 Report

The manuscript deals with the in-house developed piezoresistive pressure sensor using the SiC polytypes. It demonstrates in depth material characterisation and also presented the pressure response of the sensor at elevated temperature. As per the paper linked here (https://link.springer.com/article/10.1007/s00542-016-3102-1), the change in the sensitivity of Si based pressure sensors were estimated~ 20 % at 200 degC, however in the current manuscript the ~50 % change (Figure 14b, blue curve) in the gauge factor has been presented up to 200 degC (Sensitivity is directly proportional to the gauge factor). Since, the current manuscript utilises the SiC based piezoresistive pressure sensor, it is accepted to have a less temperature sensitivity in the pressure response. I hope the reason can be justified in the modified version of manuscript before accepting for publication.  

Author Response

Please see the attachment (we used following color-coding: in black - main parts of reply, in red - parts of your review, in green fragments of revised version of paper)  

Reviewer 2 Report

 Can write the materials and methods again to be most explicit 
The XRD discussion doesn’t show anything; please, include the reflections and JCPDS for the systems and the different patterns. 
The Raman is a similar situation that does not show the peaks, needs more discussion for all peaks, and includes the difference between systems. 
Include more details about doping and sensing. 

Author Response

(The authors gave the same response as above.)

Reviewer 3 Report

Dear Authors,

You have provided a truly up-to-date work for the present! I was glad to read and analyze your publication. I had several questions and suggestions, which I described in the file "sensors-1363227 (Review)". I hope for your early reply.

I wish you good luck and success with this and other publications in this journal!

Best wishes,

Reviewer!

Author Response

(The authors gave the same response as above.)
